# The Growth Medium Affects the Viability of IPEC-J2 Animal Cell Line in the Presence of Probiotic Bacteria

**Marie-Josée Lemay, Yves Raymond, Claude P. Champagne * and Julie Brassard ***

Saint-Hyacinthe Research and Development Centre, Agriculture and Agri-Food Canada, 3600 Casavant Blvd West, Saint-Hyacinthe, QC J2S 8E3, Canada
* Correspondence: claude.champagne@agr.gc.ca (C.P.C.); julie.brassard@agr.gc.ca (J.B.);
  Tel.: +1-450-768-7942 (J.B.)

**Abstract:** Background: The IPEC-J2 cell line is frequently used as an in vitro model to study the bioactivity of live probiotics. However, lactic acid bacteria (LB) acidify the medium, and the impact of pH and lactic acid accumulation on cell viability seem to be underestimated. Methods: IPEC-J2 viability was assessed by neutral red and flow cytometry in the presence of eight probiotics at concentrations between $10^6$ and $10^9$ bacteria/mL in maintenance and buffered media. Results: It was shown that a high inoculation level led to higher cytotoxic effects on IPEC-J2 cells after 22 h of incubation and that viability losses were more related to a combination of low pH and lactic acid than to the probiotics themselves. Furthermore, with LB at $10^6$ and $10^7$ bacteria/mL, the addition of phosphates to the media significantly reduced the drop in the pH and preserved the IPEC-J2 viability between 100% and 69%, compared to a highly variable viability between 100% and 17.5% in the unbuffered media. Conclusions: Under certain in vitro conditions, probiotics can lead to the deterioration of animal cells, and pH neutralization is an essential parameter in the cell–probiotic system in order to preserve cell viability and to better evaluate the bioactive properties of live probiotics.

**Keywords:** probiotics; IPEC-J2 cells; cytotoxicity; growth medium; buffering capacity





## 1. Introduction

In vivo tests on human or animal models are the best way to study the health effects of probiotics, but these models are complex and expensive, and the number of trials must be limited if they are used. Animal cell cultures are used for in vitro methodology for studying infection mechanisms in animal cells in substitution of the animals themselves. Recently, a cell line from the jejunum epithelium isolated from a neonatal unsuckled piglet, IPEC-J2, was characterized and used as an in vitro model system for studying porcine intestinal host–pathogen interactions and innate immune responses [1–4]. IPEC-J2 is a non-transformed, non-tumorigenic epithelial cell line with the ability to secrete mucin, produce cytokines and chemokines and express Toll-like receptors. These properties make it a unique and advantageous model, since the cells maintain their differentiated epithelial characteristics, exhibit strong similarities to the primary intestinal tissue and can simulate the functions of the innate immune response [3,4]. The use of the IPEC-J2 model could therefore prove to be very useful for screening probiotic strains and evaluating their cytotoxicity on intestinal epithelial cells.

A Scopus analysis of the literature revealed that there are at over 1300 in vitro studies using cell cultures and probiotics. Many cell cultures served in these interactions with probiotics, and at least 170 involved the CACO-2 strain alone. Therefore, a focus was required in this literature review, and it was decided to concentrate on data from IPEC-J2 cells, which were those used in this study. IPEC-J2 has been exposed to probiotics to study various immune reactions such as intestinal inflammation or cytokine production (Table S1; [2,5–10]). IPEC-J2 cell cultures have also served to study the interactions between

intestinal cells and *Salmonella* or *Escherichia* pathogens (Table S1; [11–23]), as well as with stomatitis or rotaviruses (Table S1; [2,24,25]). In more than 90% of publications where probiotic bacteria were used with IPEC-J2 cells, the goal was to study bacterial adhesion and/or prevent infection by pathogens (Table S1).

Studies investigating the effects of probiotics on the protection of animal cell cultures can be conducted using cell-free extracts, or by adding the live bacteria to the animal cell culture. It should be noted that many probiotics consist of lactic acid bacteria, which acidify their growth medium. Consequently, it is to be expected that acidification occurs during the co-incubation of probiotics and animal cell cultures. However, in 21 of the 23 studies based on interactions between IPEC-J2 cells and probiotics, the pH levels of the incubated cell cultures are not reported (Table S1). Such a lack of information also exists with respect to the evolution of cell numbers of probiotics during the incubation (Table S1). These observations drawn from the literature raise some important questions: (1) To what extent are the immune reactions of animal cell cultures influenced by the pH itself rather than by bioactive compounds (postbiotics) produced by probiotics? (2) How can preliminary assays be designed to create conditions that prevent excessive acidification/cytotoxicity? There are no data about the effect of pH/lactic acid itself on the viability of IPEC-J2 cells, nor on their interactions with bacterial and viral pathogens. However, there is evidence of cytotoxicity when the levels of probiotics are too high (Table S1; [11,21,25]). Although this situation was noted for the limited IPEC-J2 studies (Table S1), it might also be generalized for the whole sector of "probiotic–animal cell culture" experiments.

In food fermentations with lactic or probiotic cultures, the buffering capacity of the medium is a critical parameter affecting growth [26]. Media used for the extended growth of lactic cultures contain organic (citrate) or mineral (phosphate) buffers [27]. A recent study from our group [24] used such a buffered medium. However, it is not known whether changes in the buffering capacity of media that are traditionally used for animal cell cultures will affect the growth and acidification of probiotics, the viability of animal cells or their various physiological responses to probiotics.

Therefore, the goal of this study was to evaluate the cytotoxic effect of eight probiotic cultures on IPEC-J2 cells as a function of their inoculation level and the buffering capacity of the medium. The growth of the bacteria and the changes in pH were also assessed.

## 2. Materials and Methods

### 2.1. IPEC-J2 Cells

The non-transformed porcine jejunal intestinal cell line IPEC-J2 was obtained from DSMZ (ACC-701, Braunschweig, Germany). The cells were grown as described by Liu et al., 2010 [21] in Dulbecco's modified Eagle medium (DMEM/Ham's F-12 mixture, Wisent, Boucherville, QC, Canada) supplemented with 5% fetal serum bovine (FSB), 1% penicillin–streptomycin (Pen-Strep, Wisent, Boucherville, QC, Canada), 5 µg/mL insulin, 5 ng/mL transferrin, 5 ng/mL selenium solution (ITS, Corning, Ville, Pays), 5 ng/mL epidermal growth factor (EGF, Wisent, Boucherville, QC, Canada) and 15 mM HEPES (Wisent, Boucherville, QC, Canada) and maintained in a humidified atmosphere with 5% $CO_2$ at 37 °C. This multi-component medium will be referred to as the "growth medium" (GM). Cells were transferred every 3 or 4 days using 0.05% trypsin-EDTA (Wisent, Boucherville, QC, Canada) upon reaching about 90% confluency. After a minimum of 3 passages, cells were seeded on a 96-well microplate (Corning Incorporated, Kennebunk, ME, USA). For the microplate assays, a "maintenance medium" (MM) was used, which had the same composition as the GM except that it was not supplemented with FSB, penicillin–streptomycin, ITS or EGF. A buffered maintenance medium (BMM) was also prepared, which was composed of DMEM/F12 with 15 mM HEPES (MM) supplemented with 10.6 mM of $NaH_2PO_4$ (Anachemia, VWR, Mississauga, ON, Canada) and 11.8 mM of $Na_2HPO_4$ (EMD Chemicals, Merck Canada, Kirkland, QC, Canada). MM and BMM were filter sterilized (0.45 µM pores). The final pH was 7.45 ± 0.1 for MM and 7.2 ± 0.1 for BMM.

## 2.2. Bacterial Strains

The new taxonomy of lactobacilli is used in this manuscript [28]. Experiments were carried out with *Lacticaseibacillus rhamnosus* R0011, *Bifidobacterium longum* R0175, (Lallemand Health solutions, Montreal, QC, Canada), *Lactiplantibacillus plantarum* 299v (Probi, Lund, Sweden), *Lacticaseibacillus paracasei* A234, *Bifidobacterium lactis* A026, *Lactobacillus gasseri* A237 (Biena, St-Hyacinthe, QC, Canada) and a strain isolated from a commercial product containing *Lacticaseibacillus rhamnosus* GG. These strains will collectively be referred to as the "lactobacilli and bifidobacteria" (LB) group. These strains were selected because the species have recognized antiviral activities [24,29–31]. Furthermore, they are all commercially available. In most studies with *Escherichia coli*, this species is considered detrimental to the animal cells. However, there are now probiotic *E. coli* strains. Thus, this study included *E. coli* DH5$\alpha$ (Invitrogen, ThermoFisher Scientific, Burlington, ON, Canada), a containment level 1 strain (non-probiotic), as well as the probiotic *E. coli* Nissle 1917, also known as Mutaflor probiotic (FeelGood Natural Health Store, Whitby, ON, Canada).

Stock cultures of LB strains were obtained by mixing de Man, Rogosa and Sharpe (MRS)-grown (Difco, Detroit MI, USA) bacterial suspensions with sterile MRS (Difco) containing 15% ($w/v$) glycerol (Sigma-Aldrich, St. Louis, MO, USA) in a 1:5 ratio. The cell suspensions were then distributed in 1 mL cryovials (Nalgene, Rochester, NY, USA) and were frozen at $-80\ ^\circ$C. The same procedure was followed for *E. coli* cultures except that trypticase soy-broth (TSB) (Difco) was used instead of MRS. Fresh liquid inocula of LB strains were prepared by adding 1 mL of a thawed stock culture to 100 mL of MRS-AC medium and incubated aerobically at $37\ ^\circ$C until a pH of 4.5 was reached. This resulted in populations varying between 1.2 and $2.4 \times 10^9$ bacteria/mL. When the viable bacterial population is estimated by flow cytometry (Section 2.4), the data are presented in "bacteria/mL", while the results from plating on MRS agar are referred to as "CFU/mL". The MRS-AC medium was prepared by adding 1 mL of a filter-sterilized solution of 10% ($w/v$) ascorbic acid (Sigma-Aldrich, St. Louis, MO, USA) and 5% ($w/v$) L-cysteine (Sigma-Aldrich, St. Louis, MO, USA) to 100 mL of sterile MRS. For the *E. coli* inocula, 1 mL of thawed stock culture was added to 100 mL of TSB medium and was incubated aerobically at $37\ ^\circ$C for 20 h. The fresh bacterial cultures were centrifuged for 15 min at $10{,}000 \times g$ with a Beckman centrifuge (Model J-20 XPI, Rotor JLA 10.5, Palo Alto, CA, USA). The pellets were washed twice with either MM or BMM. Subsequently, the cell pellets were resuspended in MM or BMM in order to obtain bacterial cell suspensions having $1.6 \times 10^7$, $1.6 \times 10^8$ and $1.6 \times 10^9$ bacteria/mL.

## 2.3. Addition of Probiotics and Bacteria to Monolayer of IPEC-J2 Cells

IPEC-J2 cells were seeded into a 96-well microplate at 2 to $3 \times 10^4$ cells per well in GM. The GM was changed every other day. The cells formed a confluent monolayer after 4 or 5 days ($8 \times 10^4$ to $1.2 \times 10^5$ cells/well) and were then used in experiments (Figure 1). On the day before the experiment, the GM for IPEC-J2 cells was replaced with MM. After a 24 h incubation period, the culture medium was gently removed and 180 µL of either MM or BMM was added to the wells. Bacterial suspensions ($10^7$, $10^8$ and $10^9$ bacteria/mL) in corresponding MM or BMM were added at a concentration of 10% (20 µL/well) in 4 wells for each final concentration of $10^6$, $10^7$ and $10^8$ bacteria/mL. For the $10^9$ bacteria/mL concentration, 200 µL of the bacterial suspension prepared in MM or BMM at $10^9$ was added directly to the wells without any medium, but with IPEC-J2 cells.

Two controls without bacteria were required for the neutral red analyses: a positive control for IPEC-J2 viability in MM and BMM, which was considered 100% viability, and a negative control prepared by adding 10% ethanol to kill the cells. The plates were then incubated at $37\ ^\circ$C with 5% $CO_2$ for 22 h. Experiments were independently repeated 4 times. The pH was measured after the incubation period using a micro glass electrode (Accumet 13620850 model, Fisher Scientific, Toronto, ON, Canada).

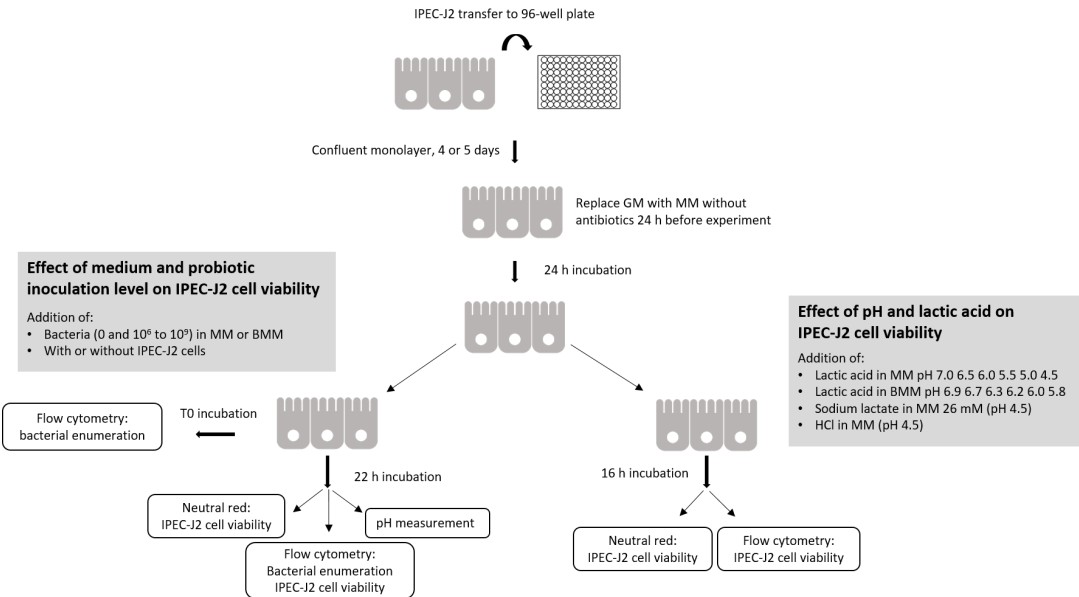

**Figure 1.** Sequence of experimental activities and analyses.

### 2.4. Enumeration of Probiotic Bacteria

The medium containing free bacteria was removed gently from each well and set aside. The IPEC-J2 monolayers were then washed twice with 200 μL of D-PBS (Wisent) and subsequently trypsinized with 25 μL of 0.05% trypsin and 0.53 mM EDTA (Wisent) for 7 min. The enzymatic action of trypsin-EDTA was stopped by adding 175 μL of GM without antibiotics. For each sample, the medium containing free bacteria, the wash solutions and the suspension containing the trypsinized cells were pooled in the same tube. All the samples were diluted in 8.5 g/L of NaCl to obtain approximately $1 \times 10^6$ bacteria/mL, which were then stained by adding 1.5 μL each of SYTO 9 and propidium iodide to 1 mL of sample and incubated in the dark for 15 min. Unless otherwise stated, the number of viable bacteria was assessed by flow cytometry using the Live/Dead BacLight Kit (Molecular Probes Inc., Eugene, OR, USA) as described by Raymond and Champagne [32]. In summary, the flow cytometry analysis was done on a CytoFlex S Flow Cytometer (Beckman Coulter, Pasadena, CA, USA). Data on cells were first acquired with the forward side scatter (FSC) (laser 488 nm) and the violet side scatter detector (VSSC) (405 nm). Additionally, the excitation of SYTO 9 was achieved with a blue laser (488 nm) and the emission was read on the FITC detector (525/40); the excitation of propidium iodide (PI) was achieved using a yellow laser (561 nm) and the emission was read on the ECD detector (610/20). Samples were run at a slow speed (10 μL/min) for 3 min. Data on cells were acquired using a threshold of 1000 on the FITC detector and 1000 on the ECD detector to eliminate noise. A region of interest (ROI) was first applied on a VSSC versus FSC dot plot (cells not colored) and used on a PI versus SYTO 9 dot plot, where the ROI for live cells was determined. The results are expressed as "flow cytometry viable" ($FC_V$) cell counts. Data analysis was performed using the Beckman Coulter Kaluza software. In order to assess the accuracy of the FCv data, traditional CFU analyses were performed on the inocula populations, using the method described by Raymond and Champagne [32], in accordance with the recommended CFU methodologies for probiotics [33].

### 2.5. Cytotoxicity Assays with IPEC-J2 Cells

Two sets of analyses were carried out on the cell viability of IPEC-J2 cultures after the 22 h incubation: neutral red uptake and apoptosis on flow cytometry (Figure 1). The neutral red uptake viability test was carried out as described by Repetto et al. [34]. Briefly, after the bacterial suspensions and control media with only MM and BMM were removed following the 22 h incubation period, 100 μL of neutral red (Sigma, 40 μg/mL in DMEM/F12 without

phenol red) prewarmed to 37 °C was added to each well containing the IPEC-J2 cells. The plate was incubated for an additional 2 h at 37 °C. The cells were then washed with 150 µL of D-PBS (Wisent), and 150 µL of extracting solution (ethanol/Milli-Q water/acetic acid, 50/49/1 (*v/v/v*)) was added to each well. The plate was shaken for 10 min. The optical density of the neutral red extract was measured using a Synergy H1 spectrofluorometer microplate reader (BioTek Instruments, Winooski, VT, USA) with excitation and emission wavelengths of 530 and 645 nm, respectively. In this test, the neutral red is absorbed by viable cells and binds to lysosomes. A loss of viability is evidenced by reduced coloration levels of cell suspensions. The percentage of viable cells was calculated using the formula:

$$\% \; viability = 100\% \times \frac{a - b}{c - b}$$

where:

a = OD derived from the wells incubated with probiotics;
b = OD derived from blank wells;
c = OD derived from the positive control.

The viability of the IPEC-J2 cells evaluated by flow cytometry was obtained by measuring apoptosis using Annexin V, Alexa Fluor™ 647 conjugate and propidium iodide (Life Technologies, Carlsbad, CA, USA). The results are expressed as "flow cytometry apoptosis" ($FC_A$). Briefly, the medium overlying the cell layer, which contained the bacteria, was gently removed from each well and kept in separate tubes. The IPEC-J2 monolayers were then washed twice with 200 µL of D-PBS (Wisent) and the wash solutions were added to the previously recovered supernatants. Cell monolayers were trypsinized with 25 µL of 0.05% trypsin and 0.53 mM EDTA (Wisent) for 7 min. The enzymatic action of trypsin-EDTA was stopped by adding 175 µL of GM, and the cell suspension was pooled with the supernatant and washed in a single tube. The IPEC-J2 cells were recovered by centrifuging the tubes at 500× *g* for 10 min using an Eppendorf centrifuge (model 5418 R, FA 45-18-11 rotor). The IPEC-J2 cell pellets were suspended in 200 µL of DMEM/F12 (Wisent) supplemented with 5% fetal serum bovine (FSB, Wisent) and then tested for apoptosis. Briefly, 20 µL of Annexin V buffer concentrated 5 times (BB5X) was added to 80 µL of sample with 5 µL of Annexin V and 2 µL of propidium iodide (1 mg/mL) and incubated in the dark for 30 min; then, 400 µL of Annexin V buffer (BB) was added before analysis. The flow cytometry analysis was performed on a CytoFlex S Flow Cytometer (Beckman Coulter, Pasadena, CA, USA). Data on the cells were acquired using a threshold of 20,000 on the SSC detector and 10,000 on the FSC detector to eliminate noise. The excitation of Annexin V647 was achieved with a red laser (640 nm) and the emission was read using the APC detector (660/20); the excitation of PI was achieved with a yellow laser (561 nm) and the emission was read with the ECD detector (610/20). Samples were run at a medium speed (30 µL/min) for 6 min. A region of interest (ROI) was first applied on an SSC versus FSC dot plot (cells not colored) and used on a PI versus Annexin V 647 dot plot, where the ROI for apoptosis was determined.

*2.6. Effect of Lactic Acid, Sodium Lactate and HCL on IPEC-J2 Viability*

The IPEC-J2 cells were grown as described above and formed a confluent monolayer after 4 or 5 days ($8 \times 10^4$–$1.2 \times 10^5$ cells per well). The acidic MM was prepared by adding lactic acid (Fisher Scientific, Toronto, ON, Canada) at a concentration of 0.2, 0.5, 0.9, 1.1, 1.3 and 1.5 g/L in order to obtain a pH of 7.0, 6.5, 6.0, 5.5, 5.0 and 4.5, respectively. The same quantity of lactic acid was added to the BMM as well, but the pH values reached were 6.9, 6.7, 6.3, 6.2, 6.0 and 5.8, respectively. Twenty-six mM of sodium lactate was added to the MM, which represented the same quantity as the quantity of lactic acid used to obtain a pH of 4.5. Finally, in one series of assays, HCl was added to the MM instead of lactic acid in order to obtain a pH of 4.5 (Figure 1). On the day of the experiment, the culture medium was removed gently from the plate and 200 µL of the MM or the BMM containing the

acidified media (acid lactic or HCl) or the sodium lactate was added to the wells. The plate was incubated at 37 °C and 5% $CO_2$ for 16 h. Media containing the acids or sodium lactate were removed gently, 100 μL of neutral red (Sigma, 40 μg/mL in DMEM/F12 without phenol red) prewarmed to 37 °C was added to each well and the plate was incubated for an additional 2 h. The cells were then washed with 150 μL of D-PBS (Wisent), and 150 μL of extracting solution (ethanol/Milli-Q water/acetic acid, 50/49/1 ($v/v/v$)) was added to each well. The plate was shaken for 10 min and the optical density of the neutral red extract was measured using a spectrofluorometer (BioTek Instruments, (Section 2.5)).

### 2.7. Statistical Analyses

Four completely independent repetitions were carried out. Statistical analyses were performed using Sigma plot software (Systat Software, San Jose, CA, USA). A three-way analysis of variance (ANOVA) was used to determine statistical significance, with $p \leq 0.05$ indicating a significant difference.

### 3. Results and Discussion

Many studies have been carried out to investigate the effects of probiotic bacteria on the physiology of animal cell cultures. Various compounds produced by probiotics can influence bioactivity, such as peptides, oligosaccharides and enzymes [35,36]. The basic premise of studies using probiotics is that they will not have a detrimental effect on the viability of the animal cells they are intended to protect. However, in excessive amounts, probiotics can cause a loss of viability of animal cells [11]. In addition, as shown by our review of the literature (Table S1), very few studies have included an evaluation of the cytotoxicity of probiotics on animal cells.

It must be kept in mind that blood and gastrointestinal in vivo conditions are basically a controlled pH environment [37]. Following the gastric transit, the pH of food is raised above 6.0 in the duodenum [38]; therefore, to ensure realistic in vitro conditions, the pH of the medium used in the animal cell cultures in the presence of probiotics must remain in a range that is representative of gastrointestinal conditions. As mentioned, a review showed that the pH of a medium itself will influence the physiology of animal cells [39]. The literature reveals that very few studies have included an analysis of the pH at the end of the incubation period with probiotics.

The results obtained in this study in relation to the pH of the medium (Figure 2) and IPEC-J2 viability (Figure 3) show sharp differences in the patterns between the two *E. coli* strains and the strains used in the LB cultures. Thus, to better ascertain the effects of comparable strains, the statistical analysis of results was carried out separately on the two *E. coli* strains and the seven LB strains (Table 1).

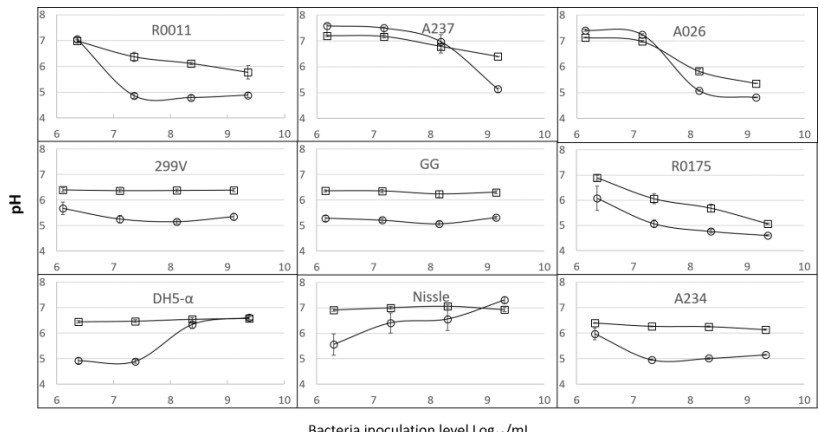

**Figure 2.** Effect of strain, inoculation level and growth medium on the pH of the media after 22 h of incubation. (○) Maintenance medium—MM; (□) buffered maintenance medium—BMM. Error bars represent the standard error of the mean.

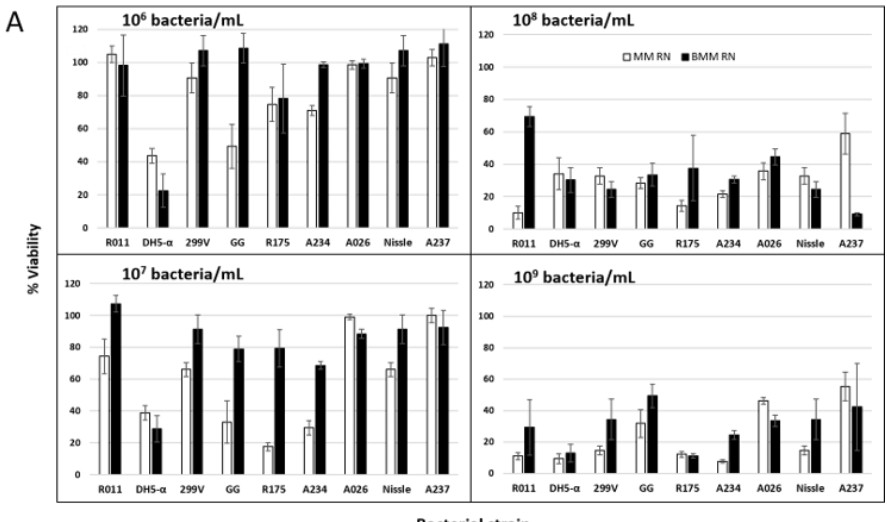

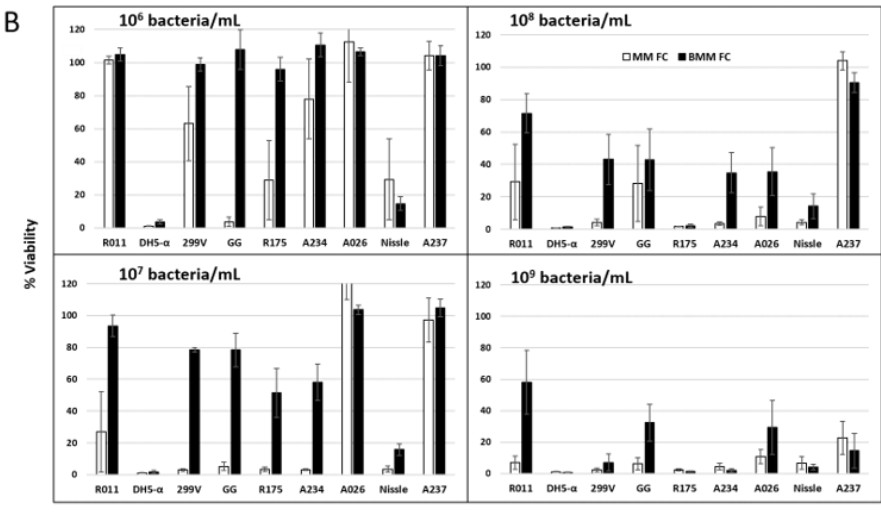

**Figure 3.** Effect of the inoculation level of 9 probiotic cultures on the viability of IPEC-2 cells after 22 h of incubation at 37 °C in the maintenance medium (MM) or the buffered maintenance medium (BMM). Panel (**A**): Results based on the neutral red assay. Panel (**B**): Results based on the flow cytometry assay. Error bars represent the standard error of the means.

**Table 1.** Statistical analyses (ANOVA) of the effect of three experimental parameters on the pH of the incubated media and on the viability levels of IPEC-J2 cells (neutral red test) in the media.

| Parameter | Probability ($p$) of a Null Effect on | | | |
|---|---|---|---|---|
| | pH of Medium | | IPEC-J2 Viability | |
| | LB [1] | *E. coli* [2] | LB | *E. coli* |
| Strain | <0.01 | <0.01 | <0.01 | 0.01 |
| Inoculation level | <0.01 | <0.01 | <0.01 | <0.01 |
| Medium | <0.01 | <0.01 | <0.01 | <0.01 |
| Strain × inoculation level | <0.01 | 0.31 | 0.01 | 0.51 |
| Strain × medium | <0.01 | 0.34 | <0.01 | 0.04 |
| Medium × inoculation level | <0.01 | <0.01 | 0.01 | 0.02 |
| Strain × medium × inoculation level | <0.01 | 0.19 | <0.01 | 0.53 |

[1] LB = Data limited to the seven *Lactobacillus* and *Bifidobacterium* cultures. [2] *E. coli* = Data limited to the two *E. coli* strains.

### 3.1. Growth and Acidification of the Media by the Lactobacilli and Bifidobacteria (LB) Cultures

The LB cultures added to the BMM without IPEC-J2 cells showed growth over the 22 h incubation period. It is noteworthy that more extensive growth of the probiotics was noted when IPEC-J2 cells were present (Table S2). At the inoculation level of $10^7$ bacteria/mL ($p = 0.002$), the value for log bacteria/mL was, on average, 0.7 higher when animal cells were present. No additional experiments were carried out to explain this occurrence. However, two hypotheses can be presented. First, in aerobic environments, lactic acid can be assimilated by animal cells, which would result in a slower drop in pH during the fermentation. Secondly, growing animal cells consume oxygen in the medium, and this could result in a lower redox level. Probiotic bacteria grow better in environments with low redox values [40].

It is notable that the bifidobacteria did not show extensive growth in the BMM. Two cultures (A237 and A026) actually showed lower viable counts by flow cytometry ($FC_V$) after incubation than the inoculated bacteria. At the inoculation level of $10^8$ bacteria/mL, the benefit of the presence of the IPEC-J2 cells was smaller—0.2 log bacteria/mL—but it was not statistically significant ($p = 0.18$). To our knowledge, this is the first demonstration that IPEC-J2 cells promote the growth of probiotics in co-cultures.

It is well known in the food industry that the bacterial growth of probiotics from LB species leads to the production of lactic acid and a decrease in pH in the medium. This could affect the viability of IPEC-J2 cells [11]. For this reason, the pH of MM and BMM was measured after 22 h of incubation of the bacteria with IPEC-J2 cells (Figure 2). The LB cultures used in this study acidified the media, but there were significant variations among the strains (Figure 2; Table 1). However, the higher the inoculation level, the greater the effect on pH in the MM and BMM fermented media; this effect was statistically significant. At the low inoculation level of about $10^6$ bacteria/mL, the pH of fermented MM containing the R0011, A237 and A026 strains remained above 7.0. The same was observed at a level of $10^7$ bacteria/mL with A026 and A237.

DMEM/F12 medium is the most widely used medium for the preparation of IPEC-J2 cell lines (Table S1). Surprisingly, no research teams, to our knowledge, have added buffers to the DMEM/F12 medium traditionally used for IPEC-J2, with or without probiotics. In this study, over 20 mM of phosphates were added to the MM (DMEM/F12 base) to create the BMM. The addition of buffers to the MM had a statistically significant impact on the final pH (Table 1). Specifically, much smaller drops in pH were noted in the BMM than in the MM in the LB cultures (Figure 2). An interaction between strain and medium (Table 1) was observed, which showed that the benefit of buffering in terms of reducing pH variations varied among the LB strains.

The acidification caused by LB cultures is presumably linked to the production of organic acids resulting from carbohydrate catabolism, and may occur despite the aerobic conditions of the experiment [41]. Strong acidification by LB bacteria is generally linked to bacterial growth [42], and this was observed following incubation (Table S2 and Figure 2). With the LB strains, a correlation was found between the log values of the $FC_V$ counts in the incubated media and the final pH ($p = 0.02$; R = 0.62). In both media, the higher the final population, the lower the pH. The strong variations observed among the strains in terms of changes to the pH of MM or BMM suggested uneven bacterial growth. As a rule, only small changes in the $FC_V$ counts were observed when the inoculation level was $10^9$ cells/mL. In this particular case, since the pH often dropped below 6.0, presumably very rapidly, uncoupling between growth and acidification occurred, which is a well-known phenomenon in lactic cultures [42].

### 3.2. Growth and Acidification of the Media by E. coli Strains

The two *E. coli* cultures grew to more than $10^9$ bacteria/mL when inoculated at a level of $10^7$ bacteria/mL or greater (Table S2), which is much higher growth than for any of the LB cultures. At low inoculation rates, the two *E. coli* strains showed between 10 and 200 times more growth than the LB cultures. However, the acidification behavior of the *E. coli* strains,

as a function of inoculation level, showed an opposite pattern to that of the LB cultures. At the low inoculation level of $10^6$ bacteria/mL, both *E.coli* DH5-$\alpha$ and Nissle produced acidic values after the 22 h incubation, while high inoculation levels led to a more neutral pH (Figure 2). *E. coli* has been shown to acidify media in anaerobic conditions [43]. Like many other bacteria, *E. coli* can utilize a variety of carbon and nitrogen sources for its growth. On excess glucose, under aerobic conditions, *E. coli* can form acidic by-products, of which acetate is the most predominant [44–54]. The production of acidic by-products is a major factor limiting high cell density growth [44,45,51,52,55]. With respect to an increase in pH, most *E. coli* cultures reutilize acetate by an activated tricarboxylic acid cycle [44,45,48,56], and they can also alkalinize the environment if amino acids are present [47]. In the present study, it can be presumed that, at the inoculation levels of $10^8$ and $10^9$ bacteria/mL, the pH dropped at the beginning of the fermentation and subsequently increased at the end, but this remains to be ascertained. Thus, even if the pH eventually returned to 7.0, the cells would have been subjected to temporary acid stress. More data are needed on the evolution of pH in the MM throughout the incubation period.

### 3.3. Effect of Probiotic Inoculation Level on IPEC-J2 Cell Viability as Ascertained by Neutral Red

This study was initiated in order to identify probiotic cultures that have anti-viral activity [24], using the viability of animal cell cultures as indicators of viral activity. Therefore, when selecting probiotics that could have anti-viral properties as detected by animal cell cultures, it is critical to ensure that the probiotic strains tested do not have detrimental effects on the animal cell cultures themselves. In the hope of obtaining the greatest protective effect from the probiotic, it is desirable to ascertain the maximum level of bacteria that can be used in the assays. The concentration of probiotic cells is an important factor to consider with a view to generate bioactivity in the bacteria [57]. Botik et al. [2] concluded that at least $10^5$ bacteria/mL of probiotics and lactic acid bacteria is needed to exert an antiviral effect on the VSV virus, with a maximal effect obtained at a concentration of $10^8$ bacteria/mL. Probiotics must be present in sufficient concentrations to produce an effect, but higher concentrations, which can lead to cytotoxicity, should be avoided.

Therefore, eight probiotics, including *E. coli* Nissle and a generic non-probiotic *E. coli*, were tested for their cytotoxicity on IPEC-J2 cells using the neutral red assay. Morcillo et al. [58] compared this method to other colorimetric methods and determined that neutral red uptake was a more sensitive method than 3-(4,dimethylthiazol-2-yl)-2,5-diphenyltetrazolium bromide assay (MTT), crystal violet and lactate dehydrogenase (LDH) analyses. The control assays performed without probiotics were used to represent 100% viability of the IPEC-J2 cells. A negative control was developed in which less than 5% viability was obtained in the ethanol-treated cells.

As was conducted with the pH data, separate statistical analyses were performed on the seven LB strains and the two *E. coli* cultures for the animal cell culture viability assays (Table 1). The results showed a significant effect of probiotic inoculation level on the viability of IPEC-J2 cells, for both the LB cultures and the *E. coli* strains. A bacterial strain effect was observed, and some cultures were found to be less toxic to IPEC-J2 cells than others (Figure 3A). At the low LB inoculation level of $10^6$ bacteria/mL in MM, 90% IPEC-J2 cell viability was maintained with four of the seven bacteria tested. At an inoculation level of $10^7$ bacteria/mL, only two bacterial strains enabled 90% cell viability after incubation (Figure 3A). However, at inoculation levels of $10^8$ to $10^9$ bacteria/mL, all probiotic bacteria generated decreases in IPEC-J2 cell viability in the MM. A correlation ($p < 0.01$; R = 0.78) was found between the bacterial populations reached after a 22 h incubation (FCv) and the IPEC-J2 viability counts as determined by the neutral red test. The R coefficient was highest with the plot at the second order (Figure S1). The data suggest that IPEC-J2 viabilities remain above 90% when LB probiotic FCv cell counts in the fermented media are at $10^8$ bacteria/mL or below (Figure S1). This is in agreement with a study by Liu et al. [25], who found that higher concentrations of *L. acidophilus* NCFM at $1 \times 10^9$ bacteria/mL and *L. rhamnosus* GG ATCC 53103 at $1 \times 10^7$ bacteria/mL significantly damaged IPEC-J2 cell

monolayers, due to reductions in the pH of the media over the 22 h incubation period. Lower concentrations, i.e., $1 \times 10^8$ bacteria/mL for *L. acidophilus* and $1 \times 10^6$ bacteria/mL for *L. rhamnosus*, were subsequently selected to preserve IPEC-J2 cell viability.

In contrast, experiments conducted by Wu et al. [12] did not show any detrimental effect of the probiotic *L. plantarum* CGMCC1258 on the IPEC-J2 cell monolayer at a level of $1 \times 10^8$ bacteria/mL. No significant change in the transepithelial electrical resistance (TEER) value was observed with this probiotic, but the contact time between the bacteria and the cells was only 6 h. Similar results were observed for various probiotics with a very short contact time of 90 min at a bacterial concentration of $1 \times 10^8$ bacteria/mL; ten different probiotic bacteria did not show any toxicity in MA104 cells [59]. It is understandable that the shorter the contact time is between the bacteria and the cell monolayer, the lower the bacterial growth and production of lactic acid will be, which is suspected to be cytotoxic. Studies conducted by other research teams have led to similar conclusions (Liu et al. [11]; Chen et al. [60]). However, as indicated previously, the post-incubation pH is not given in more than 90% of studies involving probiotic and IPEC-J2 cultures (Table S1).

The buffering capacity of the medium also had a significant effect on acidification and IPEC-J2 cell viability. At an inoculation level of $10^8$ bacteria/mL and higher, no systematic trend was observed, but IPEC-J2 viability was often low and, in many cases, similar in the MM and BMM (Figure 3A). However, with levels of $10^6$ and particularly $10^7$ bacteria/mL of LB strains, the resulting viabilities of IPEC-J2 cells were much higher in BMM than in MM (Figure 3A). However, the situation of the LB group is complex because significant interactions were noted between the strain, the medium and the inoculation level (Table 1). This means that generalizations cannot be made. Thus, as part of the process of selecting a probiotic LB strain for animal cell culture tests and determining the highest inoculation level that can be used given the need to prevent detrimental effects on the animal cells, preliminary assays must be carried out to ascertain the best conditions for given cultures.

### 3.4. Effect of Inoculation Level of E. coli Cultures on IPEC-J2 Cell Viability

In the case of the *E. coli* cultures, both the DH5-α (non-probiotic) and Nissle (probiotic) strains generated high viability losses in IPEC-J2 for all the inoculation levels tested (Figure 3). Unlike the case for the LB group, the viability of IPEC-J2 cultures was lower in BMM than MM (Figure 3). As mentioned previously, the acidification behavior differed greatly between the *E. coli* strains and the LB cultures. At the low inoculation level of $10^6$ bacteria/mL, both *E. coli* DH5-α and Nissle produced acidic values after the 22 h incubation period. This could explain the viability losses at $10^6$ and $10^7$, at least for strain DH5-α. However, the higher inoculation levels of $10^8$ and $10^9$ led to a more neutral pH (Figure 2) at the end of the incubation. It can be presumed that acidification occurred at the beginning of the incubation, to later increase. Further data are needed on the evolution of pH in the animal cell cultures as a function of the inoculation levels of *E. coli*.

*E. coli* Nissle is a Gram–negative bacterium that is widely used as a probiotic. The beneficial effects of *E. coli* Nissle are mediated through the enhancement of intestinal barrier function [61] and moderation of inflammatory disorders [62]. Furthermore, as with other probiotics, it has antimicrobial and immunomodulatory properties, such as the inhibition of pathogenic bacterial invasion of epithelial cells [63]. It was also tested against a rotavirus in a study conducted by Kandasamy et al. [64]. Their findings suggest that *E. coli* Nissle may directly interfere with rotavirus attachment to target epithelial cells, resulting in reduced virus shedding and diarrhea in piglets. Our in vitro study showed that there was a strong cytotoxic effect on the cells, even at a low rate of inoculation, and that buffered medium did not provide benefits, contrary to what was observed for the LB probiotics. Carrying out in vitro studies with probiotic *E. coli* without generating cytotoxicity poses a challenge. It can be hypothesized that a short incubation period is an element that should be considered.

### 3.5. Determination of IPEC-J2 Cell Viability by Flow Cytometry

Assays were carried out to compare the use of the neutral red method and apoptosis counts from flow cytometry ($FC_A$) for assessing the IPEC-J2 viability results. Flow cytometry was carried out on the 14 treatments presented in Figure 3B. There was a strong correlation ($p < 0.01$; R = 0.74) between the neutral red and $FC_A$ sets of data. Not only did the two methods similarly detect the variations in cell viability, but the absolute values were also similar (Figure 3). A paired *t*-test found that the viability values obtained with the neutral red test method were 5% higher than those obtained with flow cytometry, but this difference was not statistically significant *(p = 0.06)*.

These results indicate that a single flow cytometry analysis can accurately detect viable cell counts for both the IPEC-J2 and the probiotic cultures and could replace the use of both the neutral red and CFU analyses. Furthermore, flow cytometry can provide the total bacterial population and, therefore, indicate the number of dead probiotic bacteria. These data are important, as having both viable and dead bacterial counts would enable us to ascertain whether low viable counts are linked to a limited growth of the probiotic, as is currently hypothesized for the bifidobacteria, or to strong viability losses after extensive growth. More data are needed on the viable counts of bifidobacteria in the MM and BMM with IPEC-J2 cells.

### 3.6. Effect of pH and Lactic Acid on IPEC-J2 Cell Viability

Inoculation with LB cultures resulted in variable acidification levels (Figure 2) and viability losses of the IPEC-J2 cells (Figure 3). Since probiotic cultures produce many antimicrobials (bacteriocins and peptides, $H_2O_2$, enzymes) [65–69], it is not known to what extent the pH or antimicrobials generate cytotoxicity. It was therefore decided to ascertain the specific action of lactic acid and pH. MM were acidified to pH values between 7.0 and 4.5 using lactic acid, and the effect on IPEC-J2 viability in bacteria-free acidified media was evaluated. The highest viability losses were noted when lactic acid levels produced pH values below 5.5 (Figure 4). There was a significant correlation (R = 0.80; $p = 0.02$) between the pH and cell viability in the MM. When the same levels of lactic acid were added to BMM, significantly higher pH levels were reached, and animal cell viability losses were lower (Figure 4). A paired *t*-test showed that, overall, the IPEC-J2 viability was 25% greater in BMM than in MM. An assay was carried out that involved adding sodium lactate to MM at a concentration of 1.5 g/L, which was the same quantity of lactic acid used to reach a pH of 4.5 (1.5 g/L). The MM containing 1.5 g/L of sodium lactate had a pH of 7.0. After 16 h of incubation in the lactate–MM, the IPEC-J2 cell viability remained at 100% (Table S3). Thus, the detrimental effect of acidification seems to be linked to the pH itself.

The above-mentioned experiment showed that lactate at a high pH was not cytotoxic to IPEC-J2 cells. Since a combined detrimental effect of lactate/lactic acid at a low pH was possible, a second assay was carried out in which HCl was added to the MM in order to obtain a pH of 4.5. Under those conditions, cell viability decreased to 60% following the 16 h incubation period. However, a greater decrease in viability was observed with lactic acid at a pH of 4.5 (Figure 4). Therefore, the detrimental effect of acidification on the IPEC-J2 cells is linked to both pH and the presence of organic acids. In order to better understand the effect of pH and to create conditions mimicking those found in the small intestine, buffers were added to the maintenance media (DMEM/F12). Our findings demonstrate that adding buffers (1.3 g/L of $NaH_2PO_4$ and 1.7 g/L of $Na_2HPO_4$) to the maintenance media reduced the drop in pH for most bacteria tested at the inoculation levels of $10^6$ bacteria/mL and $10^7$ bacteria/mL. At the $10^7$ bacteria/mL level, only two bacteria tested allowed high cell viability (90%). However, when the buffers were added to the media, four bacteria led to an 88% cell viability and six maintained more than 79% viability. These results led to the conclusion that low pH values are more related to the cytotoxicity observed in IPEC-J2 than the probiotics themselves. An in vitro evaluation of the cytotoxicity of probiotics should therefore be carried out under conditions similar to those present during digestion.

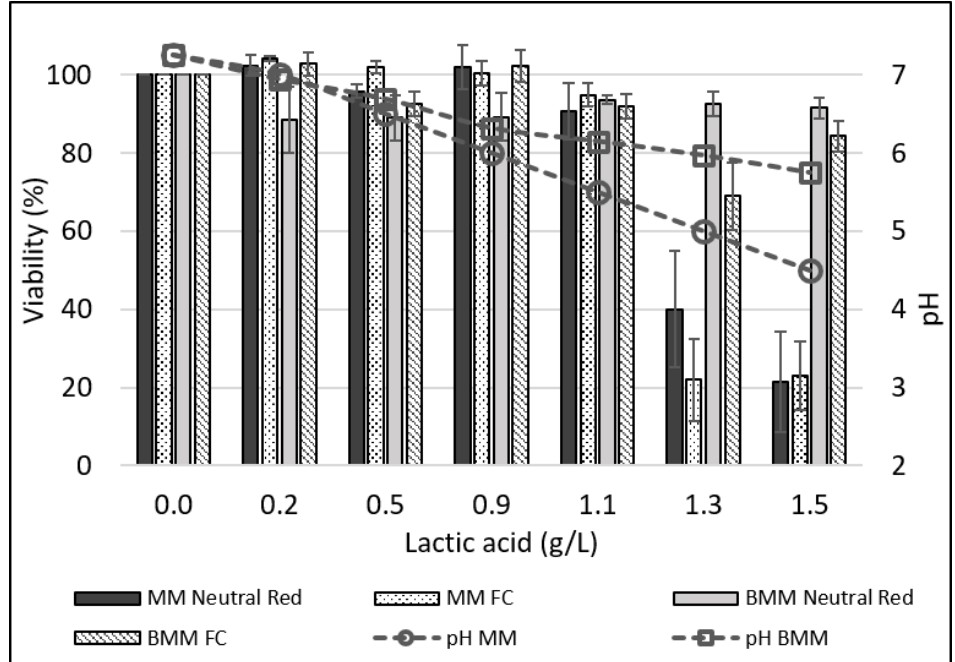

**Figure 4.** Effect of the pH adjustment of the maintenance medium (MM) or buffered maintenance medium (BMM) on IPEC-J2 cell viability (%) obtained by neutral red or apoptosis in flow cytometry (FC$_A$).

This study innovates by demonstrating that increasing the buffering capacity of the animal cell growth medium can (1) prevent undesirable acidification and/or (2) allow increased inoculation rates of probiotics under some conditions. Furthermore, the results show that the detrimental effects of probiotics on animal cell viability are linked to a combination of low pH and lactic acid.

## 4. Conclusions

This study confirms that inoculating probiotic bacteria that are deemed safe for IPEC-J2 cells can generate viability losses if the inoculation levels are too high and/or the incubation period is too long. Data from this study allow the formulation of three recommendations to research teams planning to carry out any experiment in which probiotic cultures are blended with animal cell lines in order to assess biological effects:

(1) Preliminary trials must be carried out to identify the experimental conditions allowing the highest inoculation level at which a given probiotic strain does not negatively affect the viability of the animal cell cultures; this needs to be determined for each bacterial strain.

(2) The pH of the medium as well as the bacterial counts should be registered, not just at the beginning of the experiment, but also at the end of the incubation.

(3) New media are required; increasing the buffering capacity of the traditional media for IPEC-J2 cultures reduces the drop in pH, which affects the cell viability. This allows for the use of higher inoculation levels of probiotics, which can provide health benefits for the animal cell cultures. Nevertheless, more research should be carried out on how to improve buffering of the medium for animal cell lines when probiotic cultures are inoculated.

Data from this study suggest that the antiviral activities of two of the probiotics reported in a recent publication [24] are indeed linked to bioactivity of the bacteria other than acidification of the medium. Unfortunately, this has not been widely ascertained in the past. Since the pH in the intestines is above 6.0, in vitro conditions that result in a pH value below 6.0 may not be representative of in vivo conditions.

**Supplementary Materials:** The following supporting information can be downloaded at https://www.mdpi.com/article/10.3390/applmicrobiol2040058/s1, Figure S1: The effect of the bacterial population reached in the BMM after the 22 h incubation (FCv) and the resulting viability levels of the IPEC-J2 cells (assessed by the neutral red assay); Table S1: Data on growth and acidification of some probiotics in media with IPEC-J2 animal cell cultures; Table S2: Growth of the various probiotics in the buffered maintenance medium (BMM) at inoculation levels of approximately $10^7$ and $10^8$ cells/mL after a 22 h incubation in the presence of (cell) or absence of (no cell) IPEC-J2 cells. Values consist of viable counts in a flow cytometry analysis ($FC_V$); Table S3: Viability of IPEC-J2 in incubated maintenance medium (MM) at a pH of 5.0, as well as in a bacteria-free MM adjusted to a pH of 5.0 with lactic acid.

**Author Contributions:** Conceptualization, all authors; methodology, M.-J.L. and Y.R.; formal analysis, C.P.C., M.-J.L. and Y.R.; resources, C.P.C.; data curation, all authors.; writing—original draft preparation, M.-J.L. and C.P.C.; writing—review and editing, all authors; supervision, C.P.C.; project administration, C.P.C. and J.B.; funding acquisition, J.B. All authors have read and agreed to the published version of the manuscript.

**Funding:** This research was funded by Agriculture and Agri-Food Canada, project number J-001305.

**Data Availability Statement:** Not applicable.

**Conflicts of Interest:** The authors declare no conflict of interest.

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
