# Peer review of "The Growth Medium Affects the Viability of IPEC-J2 Animal Cell Line in the Presence of Probiotic Bacteria"

_2673-8007, doi:10.3390/applmicrobiol2040058_

Round 1
Reviewer 1 Report
Cells/cell lines grow only at optimum pH, altering the buffering system obviously results in reduced growth of cells/death in the invitro condition. For screening and testing the efficacy of probiotics, there are many simple methods are available. Need clarification in this regard.
Authors should explain on what basis the concentration of Lactic acid chosen for this experiment (0.2 to 1.5g per liter) in the manuscript text. Is there any reference for fixing the concentration of Lactic acid?
Line 16-17: Authors are asked to reframe the sentence.
Authors should include more literature in the introduction.
Authors should give the references for line 40 to 42.
Authors should explain why they have chosen this specific 9 probiotic cultures for the experiments.
Authors should elaborate/explain the abbreviation FCA (line 437 and line 439) in the manuscript text.
Again, authors should elaborate/explain the abbreviation FCD (line 448 and line 449) in the manuscript text.
Author Response
Reviewer 1
R1C: Cells/cell lines grow only at optimum pH, altering the buffering system obviously results in reduced growth of cells/death in the in vitro condition. For screening and testing the efficacy of probiotics, there are many simple methods are available. Need clarification in this regard.
R1C Authors response: The introduction has been reworked and the problematic better defined.
R1C1: Authors should explain on what basis the concentration of Lactic acid chosen for this experiment (0.2 to 1.5g per liter) in the manuscript text. Is there any reference for fixing the concentration of Lactic acid?
R1C1 Authors response: The lactic acid concentrations were chosen to achieve certain pH values. This of course varies with the buffering capacity of the media. As a result, the pH values attained differed between the two media for a given lactic acid addition level. Details have been added for this purpose around lines 237-238 of the clean version of the manuscript.
R1C2: Line 16-17: Authors are asked to reframe the sentence.
R1C2 Authors response : in reference to comment R3C1, the abstract has been completely reworded.
R1C3: Authors should include more literature in the introduction.
R1C3 Authors response: A table with data from 22 references the literature was prepared in the first version. However, such tables are typically found in review-type-articles. This explains why it was decided to present it as « Supplementary Table ». The initial strategy was to simply refer the reader to this table, but the reviewer’s comment incited us to present the highlights of all these data in the introduction. As a result the introduction was lengthened, and 15 references were cited. The potential problem of studies that did not consider acidification of the medium as a contributing factor in results was particularly emphasized. We added our own recent publication on the subject.
R1C4: Authors should give the references for line 40 to 42.
R1C4 Authors response: This section was expanded, and was part of the numerous additions that were made in response to R1C3.
R1C5: Authors should explain why they have chosen this specific 9 probiotic cultures for the experiments.
R1C5 Authors response: Three elements served in the decision to use these strains. First , the literature showed that L. rhamnosus GG had demonstrated antiviral activities (Lee et al., 2015; Liu et al., 2013), and this was also the case for other species used in this study (Rodríguez-Díaz and Monedero 2013). A second reason is that there is commercial availability of off the strains, since they can be purchased from two companies. The third reason is that we had just published a manuscript on antiviral properties of the strains (Leblanc et al. 2022) and it was necessary to show that the methodologies used were carefully designed to eliminate the potential of acidification and justify the use of a more highly buffered medium. We added this information in the methodology section in case readers would have the same interrogation as did the reviewer.
Lee, D. K., Park, J. E., Kim, M. J., Seo, J. G., Lee, J. H., & Ha, N. J. (2015). Probiotic bacteria, B.longum and L.acidophilus inhibit infection by rotavirus in vitro and decrease the duration of diarrhea in pediatric patients. Clinics and Research in Hepatology and Gastroenterology, 39, 237-244.
Liu, F., Li, G., Wen, K., Wu, S., Zhang, Y., Bui, T., Yang, X., Kocher, J., Sun, J., Jortner, B., & Yuan, L. (2013). Lactobacillus rhamnosus gg on rotavirus-induced injury of ileal epithelium in gnotobiotic pigs. Journal of Pediatric Gastroenterology and Nutrition, 57, 750-758.
Rodríguez-Díaz, J., & Monedero, V. (2013). Probiotics against digestive tract viral infections. In Bioactive Food as Dietary Interventions for Liver and Gastrointestinal Disease (pp. 271-284): Elsevier.
Leblanc, D.; Raymond, Y.; Lemay, M.J.; Champagne, C.P.; Brassard, J. Effect of probiotic bacteria on porcine rotavirus OSU infection of porcine intestinal epithelial IPEC‑J2 cells. Archives of Virology, 2022, 167(10): 1999-2010
R1C6: Authors should elaborate/explain the abbreviation FCA (line 437 and line 439) in the manuscript text.
R1C6 Authors response: The abbreviations for FCA, FCV are available in the methodology (sections 2.4 and 2.5) on lines 181, 211-212 of the clean version of the manuscript. They were repeated in sections 3.1 and 3.5
R1C7: Again, authors should elaborate/explain the abbreviation FCD (line 448 and line 449) in the manuscript text.
R1C7 Authors response: Clarifications have been made.
Reviewer 2 Report
This manuscript investigated that effect of growth medium on the viability of IPEC-J2 cells in the presence of probiotics. However, this manuscript lacks in-depth research and discussion. Also, the meaning of some sentences in this paper is not clear. In addition, it is noted that the manuscript needs careful edition of English grammar, spelling, and sentence structure. Therefore, I think that this paper should undergo major revisions before publication.
The following are the questions in this manuscript:
1. Line 18 “bacteria/ml” should be “bacteria/mL”, line 46 “However” should be “However,”, line 62 “assessed” should be “assessed.”, and line 241 “difference..” should be “difference.”, etc.
2. These sentences expression is difficult to understand and needs to be rewritten. Like lines 28-29, 40-41, 56-59, etc.
3. The whole manuscript is slightly repeated in the experimental steps, so I suggest that language should be refined.
4. Line 94-95 “However, there are now probiotic…” lacks relevant literature support.
5. There are many format problems in this manuscript that need to be corrected, like line 134-135, line 143-146, line 196-210, etc.
6. The experimental content of this paper is too simple and lack of in-depth research.
7. I think that the discussion in paragraph 3.4 does not explain the experimental phenomenon in Figure 3 well.
8. There are too few discussion parts, which should be explained in depth.
9. The conclusion of the manuscript lacks corresponding data, as shown in lines 527-530.
10. References have some details and format errors. Like line 561, line 616, etc.
Author Response
Reviewer 2:
R2C: This manuscript investigated that effect of growth medium on the viability of IPEC-J2 cells in the presence of probiotics. However, this manuscript lacks in-depth research and discussion. Also, the meaning of some sentences in this paper is not clear. In addition, it is noted that the manuscript needs careful edition of English grammar, spelling, and sentence structure. Therefore, I think that this paper should undergo major revisions before publication.
R2C Authors response: The article was reviewed by a professional translation and editing service of Agriculture and Agri-Food Canada and changes were made throughout the manuscript.
The following are the questions in this manuscript:
R2C1: Line 18 “bacteria/ml” should be “bacteria/mL”, line 46 “However” should be “However,”, line 62 “assessed” should be “assessed.”, and line 241 “difference..” should be “difference.”, etc.
R2C1 Authors response: the corrections have been made.
R2C2: These sentences expression is difficult to understand and needs to be rewritten. Like lines 28-29, 40-41, 56-59, etc.
R2C2 Authors response: In response to comment R1C3, R1C4, the 40-41, 56-59 lines of the introduction have been reworded.
R2C3: The whole manuscript is slightly repeated in the experimental steps, so I suggest that language should be refined.
R2C3 Authors response: The text was shortened to try to avoid repetitions that are not essential. The “Revision” mode of MS Word allows one to identify these cuts.
R2C4: Line 94-95 “However, there are now probiotic…” lacks relevant literature support.
R2C4 Authors response: This comment is similar to that of R1C3 and R1C4. As mentioned earlier, 15 references were cited.
R2C5: There are many format problems in this manuscript that need to be corrected, like line 134-135, line 143-146, line 196-210, etc.
R2C5 Authors response: the corrections have been made.
R2C6: The experimental content of this paper is too simple and lack of in-depth research.
R2C6 Authors response: The procedures appear simple because they are based on growth and acidification of media by lactic acid bacteria which is a common experiment, even in animal cell cultures. However, as shown in our literature review, most research teams have not examined the possibility of acidification of the media in which the animal cells are growing, and its effect of the state of the animal cells. It must be kept in mind that he pH of the small intestine and of blood are above 6.0. Thus, although it could be argued that acidification is a normal occurrence with lactic cultures this does not occur in real in vivo conditions. Most teams have not considered this, which may invalidate their results. Our data means to provide practices that would avoid acidification in the animal cell cultures and prevent experimental errors. Thus although the experimental conditions are in appearance simple, it is their impact on the sector which are notable. We even present three recommendations to improve experimental conditions in the Conclusion section. Furthermore it should be mentioned that state-of-the-art flow cytometry was used in the study, which is far from “simple”.
R2C7: I think that the discussion in paragraph 3.4 does not explain the experimental phenomenon in Figure 3 well.
R2C7 Authors response: The text was modified and an explanation was provided. It is hypothetical at inoculation levels of 108 and 109, but the need for more research is pointed out.
R2C8: There are too few discussion parts, which should be explained in depth.
R2C8 Authors response: We have added discussion elements at the beginning of the discussion as well as in sections 3.1 and 3.4
R2C9: The conclusion of the manuscript lacks corresponding data, as shown in lines 527-530.
R2C9 Authors response: The conclusion was re-written with an emphasis on recommendations directly linked to our data. We believe that these recommendations could have a significant impact on the sector as they may apply to other animal cell lines than IPEC-J2.
R2C10: References have some details and format errors. Like line 561, line 616, etc.
R2C10 Authors response: the corrections have been made.
Reviewer 3 Report
The manuscript "The Growth Medium Affects the Viability of IPEC-J2 Animal Cell Cultures in the Presence of Probiotic Bacteria" has an interesting result. However, the manuscript needs to be improved for publication in Applied Microbiology. The discussion needs to review and compare the data with the literature. Authors are suggested to check typos throughout the manuscript. Recommended checking the manuscript with a professional English Editing service.
- I find the abstract too general, which does not allow readers to understand what the authors did and what they found. Given that this part is the most important and I believe authors should reformulate the abstract to become more specific following the Journal's guidelines.
- L16, 135: 22 h
- L18: 106 and 107 bacteria/mL
- Authors are suggested to use the most updated references throughout the introduction section (last 5 years).
- Many spacing and punctuation problems are found in Table S1.
- L62: “will also be assessed”-use the full stop (.)
- L73, 135, 177: Insert space between "37" and " ºC" and make corrections throughout the manuscript.
- L74: 0.05 % trypsin-EDTA
- L76: “96 wells microplate”- Sources (manufacturer name, city, country) need to mention all the chemicals, reagents, and equipment used in this manuscript.
- L96, 97: Use the full form of bacteria name only first time throughout the manuscript (e.g., E. coli DH5α)
- L99: De Man, Rogosa and Sharpe (MRS) agar
- L100, 104, 113: If the product is from the same company, mention only the company name (e.g., sterile MRS (Difco). No need to mention the city and country name. Make corrections throughout the manuscript.
- L115, 263: 20 h
- L122: “probiotics and bacteria”-did you investigate two different bacteria?
- L142: “separate tubes for each strain and inoculation level”- meaning is not clear. Did you enumerate the population of free bacteria from each well?
- L145: 0.53 mM
- L275, 366, 444: “P” should be italic. Check and make corrections throughout the manuscript
- Figure 3: Follow the journal style for the SI unit (bacteria/mL).
- L410: Error bars represent the Standard Error of the Means.
Author Response
Reviewer 3 :
R3C: The manuscript "The Growth Medium Affects the Viability of IPEC-J2 Animal Cell Cultures in the Presence of Probiotic Bacteria" has an interesting result. However, the manuscript needs to be improved for publication in Applied Microbiology. The discussion needs to review and compare the data with the literature. Authors are suggested to check typos throughout the manuscript. Recommended checking the manuscript with a professional English Editing service.
R3C Authors response: The article was reviewed by a professional translation and editing service of Agriculture and Agri-Food Canada and changes were made throughout the manuscript. Several format corrections were made and clarifications and comparisons with the literature were made in the discussion (see responses to comments R2C7, R2C8).
R3C1: I find the abstract too general, which does not allow readers to understand what the authors did and what they found. Given that this part is the most important and I believe authors should reformulate the abstract to become more specific following the Journal's guidelines.
R3C1 Authors response: the abstract has been completely reworded.
R3C2 : L16, 135: 22 h
R3C2 Authors response: the corrections have been made.
R3C3: L18: 106 and 107 bacteria/mL
R3C3 Authors response: the correction has been made.
R3C4: Authors are suggested to use the most updated references throughout the introduction section (last 5 years).
R3C4 Authors response: In connection with comments R1C3, R1C4, and R2C2, the introduction has been reworked as well as the problematic surrounding the purpose of this study.
R3C5: Many spacing and punctuation problems are found in Table S1.
R3C5 Authors response: the corrections have been made.
R3C6: L62: “will also be assessed”-use the full stop (.)
R3C6 Authors response: the correction has been made.
R3C7: L73, 135, 177: Insert space between "37" and " ºC" and make corrections throughout the manuscript.
R3C7 Authors response: the corrections have been made.
R3C8 : L74: 0.05 % trypsin-EDTA
R3C8 Authors response: the correction has been made.
R3C9: L76: “96 wells microplate”- Sources (manufacturer name, city, country) need to mention all the chemicals, reagents, and equipment used in this manuscript.
R3C9 Authors response: the correction has been made.
R3C10: L96, 97: Use the full form of bacteria name only first time throughout the manuscript (e.g., E. coli DH5α)
R3C10 Authors response: the corrections have been made.
R3C11: L99: De Man, Rogosa and Sharpe (MRS) agar
R3C11 Authors response: the correction has been made.
R3C12: L100, 104, 113: If the product is from the same company, mention only the company name (e.g., sterile MRS (Difco). No need to mention the city and country name. Make corrections throughout the manuscript.
R3C12 Authors response: the corrections have been made.
R3C13 : L115, 263: 20 h
R3C13 Authors response: the corrections have been made.
R3C14: L122: “probiotics and bacteria”-did you investigate two different bacteria?
R3C14 Authors response: The probiotics (7 lactobacilli and E. coli Nissle) included in this study were described in lines 106-113 of the clean version of the manuscript. Because E. coli Nissle has recently been recognized as a probiotic, the authors chose to also include E. coli DH5α as a control (lines 114-119 of the clean version of the manuscript). We wished to assess whether E. coli Nissle behaved more like a probiotic or more like an E. coli that refer here as the "bacteria" in the various assays.
R3C15: L142: “separate tubes for each strain and inoculation level”- meaning is not clear. Did you enumerate the population of free bacteria from each well?
R3C15 Authors response: This section has been reworked for more clarity (lines 160-166 of the clean version of the manuscript).
R3C16 : L145: 0.53 mM
R3C16 Authors response: the correction has been made.
R3C17: L275, 366, 444: “P” should be italic. Check and make corrections throughout the manuscript
R3C17 Authors response: the corrections have been made.
R3C18: Figure 3: Follow the journal style for the SI unit (bacteria/mL).
R3C18 Authors response: the correction has been made.
R3C19: L410: Error bars represent the Standard Error of the Means.
R3C19 Authors response: the correction has been made.
Round 2
Reviewer 1 Report
Authors revised the manuscript according to the comments raised by the reviewers and clearly explained in the response section. Now the revised manuscript is fine and accepted for publication.